# Diffusion and Velocity Correlations of the Phase Transitions in a System of Macroscopic Rolling Spheres

**DOI:** 10.3390/e24111684

**Published:** 2022-11-18

**Authors:** Francisco Vega Reyes, Álvaro Rodríguez-Rivas, Juan F. González-Saavedra, Miguel A. López-Castaño

**Affiliations:** 1Departamento de Física, Universidad de Extremadura, 06071 Badajoz, Spain; 2Instituto de Computación Científica Avanzada (ICCAEx), Universidad de Extremadura, 06071 Badajoz, Spain; 3Department of Physical, Chemical and Natural Systems, Pablo de Olavide University, 41013 Sevilla, Spain

**Keywords:** phase transition, diffusion, granular matter

## Abstract

We study an air-fluidized granular monolayer composed of plastic spheres which roll on a metallic grid. The air current is adjusted so that the spheres never lose contact with the grid and so that the dynamics may be regarded as pseudo two dimensional (or two dimensional, if the effects of the sphere rolling are not taken into account). We find two surprising continuous transitions, both of them displaying two coexisting phases. Moreover, in all the cases, we found the coexisting phases display a strong energy non-equipartition. In the first transition, at a weak fluidization, a glass phase coexists with a disordered fluid-like phase. In the second transition, a hexagonal crystal coexists with the fluid phase. We analyze, for these two-phase systems, the specific diffusive properties of each phase, as well as the velocity correlations. Surprisingly, we find a glass phase at a very low packing fraction and for a wide range of granular temperatures. Both phases are also characterized by strong anticorrelated velocities upon a collision. Thus, the dynamics observed for this quasi two-dimensional system unveil phase transitions with peculiar properties, very different from the predicted behavior in well-know theories for their equilibrium counterparts.

## 1. Introduction

The dynamics of granular matter has been an emerging field for several decades now [1,2]. This is in part due to the fact that this kind of material has many industrial and engineering applications [3] and is also partly due to the fact that granular setups can be used as prototype non-equilibrium systems for experiments [4] and also from a theoretical viewpoint. Thus, they allow for the development of the theory of non-equilibrium statistical mechanics [5], fluid mechanics [6], and materials science [7,8]. Moreover, advances in the theory of granular dynamics have clearly put in evidence that, at both the mesoscopic and macroscopic levels, the dynamics of granular matter can present analogous phenomenology to that of molecular matter and at the same time is usually present with a higher degree of complexity [1,5,9]. This is the case, for instance, of phenomena such as stratification [3], diffusion [10,11,12], segregation [13,14,15,16], mixing [3,17], laminar flow [6], hydrodynamic instabilities [18,19], convection [18,20,21], turbulence [22,23], jamming [24,25], memory effects [26,27], and phase transitions [17,28,29,30,31,32], just to name a few. With respect to a phase transition, granular matter displays disordered states which can be, for instance, liquid-like, glass transition or hyperuniform states, and ordered structures, such as nematic phases [33,34], hexagonal [29,35], or cubic crystals [17,36]. It also shows phases that are exclusive to two-dimensional (2D) systems, such as the hexatic phase. All of these phenomena appear in granular matter in significantly more complex forms, in comparison with molecular matter.

As an example of the higher complexity of granular dynamics, and just out of illustration, the set of the steady base flows that can be observed in a plane Fourier/Couette configuration (a fluid confined within two infinite parallel walls) includes those that are present in molecular gases plus new steady flows, which are specific to granular fluids [6]. In particular, the Fourier configuration (two static parallel walls) for a molecular gas yields steady flows with a constant heat flux; these constant heat flux states are, however, possible in a granular gas if the confining parallel walls are moving (Couette configuration) [37]. Moreover, in addition to this complex phenomenology that is also present in simpler forms in molecular matter, there are phenomena in the granular matter that do not have an analog in their equilibrium counterparts, such as granular nucleation [38], inelastic clustering [39], collapse [4], or velocity correlations that, at a low density, clearly violate the molecular chaos assumption [40]. So, granular dynamics can be regarded, from a theoretical point of view, as an extension or generalization of the dynamics of molecular matter [5,9].

In this work, we focus instead on the phase transitions and the order/disorder phenomenology in a monolayer of macroscopic spheres. Spheres are fluidized by turbulent air currents in such a way that they keep rolling over a horizontal plane at all times. In this way, the dynamics remain, after activation, as quasi-2D (or pseudo-2D, as preferred). On the other hand, and as it is well known, a 2D equilibrium fluid crystallizes to a hexagonal phase via a continuous transition that is mediated by a phase that is specific to two dimensions (the hexatic phase). This process is well-described by the KTHNY scenario (from their main authors Kösterlitz, Thouless, Halperin, Nelson, and Young; see their independent works [41,42,43,44]). The hexatic phase appears exclusively in 2D and is characterized by having quasi-long-ranged orientational correlations (with a power-law slow decay) and short-ranged translational correlations (with an exponential decay). By contrast, the hexagonal crystal shows the quasi-long-ranged translational order. Thus, as the crystal melts, the long translational order is lost, and the correlations undergo a complete transformation process toward complete disorder, which characterizes the liquid phase [45]. Moreover, the melting process is defect-mediated [45].

This liquid–hexatic–crystal scenario has been observed in non-equilibrium systems as well, although with (eventually) important differences. For instance, the hexatic phase has been detected in a monolayer of vertically vibrated macroscopic spheres [35,46]. More specifically, in the work by Olafsen and Urbach [35], the phenomenology for the quasi-2D non-equilibrium system appears to be rather similar to that described for equilibrium systems that are strictly two dimensional. In effect, a first transition was found, where the crystal melts to the intermediate hexatic phase described by the KTHNY scenario, by means of the expected (according to the predictions of the KTHNY scenario) process of the unbinding of the dislocation pairs in the hexagonal lattice. A second transition consecutively then occurs, where the hexatic phase decays to a liquid-like phase with complete disorder, through a process of the gradual unbinding of disclinations. In this way, the crystal melting process occurs as a double continuous transition, without a coexistence with the liquid phase ever taking place, unlike in three-dimensional matter. In the work by Olafsen and Urbach [35], however, experiments were performed with only stainless steel spheres, which are nearly elastic (see the work by Louge and collaborators [47], accompanied with the comprehensive data table [47], where the coefficients of the restitution for steel and other metals are given). In fact, the corresponding velocity distribution shows little to no deviations off the Maxwellian distribution [48,49], meaning that the system is not far from equilibrium [50,51]. In this way, one can say that the phase behavior might be expected not to differ much from that of a truly equilibrium system. To the point that the experiments by Olafsen and Urbach [35] in fact show virtually no difference with respect to the equilibrium theory. However, the differences between the 2D phase behavior in granular matter and the KTHNY were in fact to appear later. In this sense, further experiments performed by Komatsu and Tanaka [46], with the same monolayer configuration, found an intriguing disappearance of the KTHNY scenario. The key here is they used rubber spheres, which are more inelastic than steel spheres [47]. Thus, non-equilibrium effects are significantly stronger in their experiments, in comparison with the previous results by Olafsen and Urbach [35]. Moreover, Komatsu and Tanaka observed an abrupt change from the continuous to the discontinuous melting. In fact, the melting transition for very inelastic spheres was found to exhibit a phase coexistence between the crystal and the liquid, without the hexatic phase ever showing up in the process, i.e., the KTHNY scenario disappears completely at a high inelasticity. (In fact, a previous result for brass spheres, also more inelastic than stainless steel [47], in a two-layer system, already proved that increasing the degree of inelasticity can importantly alter the phase behavior of the granular layer [30]).

Furthermore, the 2D phase behavior in active matter seems to be even more complex than in granular matter. In fact, a complex mélange between the KTHNY scenario with the motility-induced phase separation (that is characteristic of active matter [52]) can be observed for wide ranges of particle density and at strong particle activity [53]. In summary, according to strong experimental and computational evidence, the KTHNY scenario is just one of the possible realizations of the phase behavior in 2D or nearly 2D non-equilibrium systems. Moreover, alternative scenarios in equilibrium systems seem to be posssible as well and they depend upon the type of interaction between the particulate system constituents [45]. In effect, first-order transitions in 2D equilibrium systems seem to be viable, according to the theoretical analysis and Monte Carlo simulations, if the dislocation and disclination unbinding are concurrent [45]. Moreover, this concurrence appears to depend upon the features of the interactions between the particles of the system. Therefore, particle interactions seem to play an important role in the phase behavior in 2D. This idea motivates the present work.

Thus, the present work, which deals with a system of air-fluidized ping-pong balls, is motivated by the previous discussion on the phase behavior in 2D particulate systems. A previous experimental observation puts in evidence the existence of long-range repulsive interactions between air-fluidized particles [54]. This interaction, whose origin lies in the hydrodynamic interactions due to an interstitial fluid [55] (air in this case), is absent in the case of the vibrated system. Moreover, in our case, this repulsion does not prevent particle direct encounters (collisions) [54,56]. Thus, due to the combined action of inelastic collisions plus the repulsive potential between particles, a different phase behavior could be expected. An analysis is performed of the eventual departures from the KTHNY scenario when long-ranged hydrodynamic interactions would be relevant for a more complete understanding of the phase behavior in two-dimensional systems. Although this experimental configuration had been studied in a number of previous works [54,56,57,58,59], the focus was put on other phenomenology, and no detailed analyses were carried out on the phase behavior, except for more recent work where a generic description has been provided [56,59].

For this reason, we focus in this work on the specific features of the observed phases for air-fluidized rolling spheres and the transitions between them. As we will see, several discontinuous phase transitions can clearly be detected as the air upflow intensity is increased, giving rise to states with either a single phase or two coexisting phases. We will perform the study as a function of the average granular temperature of the system (here defined as $T=(1/2m\langle v^{2}\rangle )$, where $\langle v^{2}\rangle$ is the square of the particle velocity spatially averaged over all the steady states). We will report the results for the system particle density, granular temperature field, and pair correlation function. Additionally, we study the diffusion and velocity autocorrelations of each of the observed phases, and we report a strong discontinuity for these magnitudes between the coexisting phases. Moreover, the combined analysis of these magnitudes allows us to identify the following phases: the arrest, glass, liquid, and hexagonal crystal phases. In particular, we show that the glass and crystal phases are clearly subdiffusive. Surprisingly, the liquid phase can display either normal diffusion or weakly subdiffusive (or superdiffusive) behavior. As we will see, these transitions in the diffusive behavior of the system occur in a discontinuous way. Furthermore, in the glass phase, particle velocities are strongly anticorrelated at early times, whereas the crystal anticorrelations are weak. We also found a strong energy non-equipartition in all the cases of the two coexisting phases. Most importantly, we will report the complete absence of the KTHNY scenario in all the reported results here, which comprise a comprehensive set of experiments.

The paper is structured as follows: Section 2 is devoted to the description of the experimental setup and methods and also to a qualitative description of the observed phase behavior. In Section 3, the results for the particle diffusion and velocity correlations of each of the observed phases are analyzed separately and in detail. Finally, in Section 4, the results and final conclusions are discussed.

## 2. Description of the Experiments

### 2.1. Setup

The experimental configuration we used in this work was designed in our lab. It consists of an air table setup [60]. In our case, it is composed of two essential parts: (a) the driving unit, which produces a stable quasi-laminar air upflow, consisting of a high-power fan (SODECA HCT-71-6T) coupled to a system of short tunnel winds; and (b) the arena, which consists of a flat metallic plate with a hexagonal lattice of perforatecd circular holes (of 3 mm diameter) which is surrounded by circular walls (PLA plastic) of 4.5cm height. The metallic plate is carefully leveled to be horizontal (so that gravity does not enter into the dynamics if it is restrained within the plate). Both parts are connected by a pair of perpendicular channels that conduct the air released from the fan upward to the metallic grid. See Figure 1 for a schematic representation of this configuration. A set of spherical particles (ping-pong balls, made of ABS plastic with mass density $0.08\;g\;{\mathrm{cm}}^{-3}$) are placed on the metallic grid. The spherical particles are all identical, having a diameter of σ=4cm and a mass density $\rho =0.08\;g\;{\mathrm{cm}}^{-3}$ (ABS plastic material). The metallic grid has a square shape ($80\times 80\;{\mathrm{cm}}^{2}$). A (circle-shaped) plastic wall is placed inside, centered, so that the particles are enclosed within this circular region of radius R=36.25cm.

In the middle of the conducting channels, there is a foam that homogenizes the upflow. This foam helps the upflow reach under quasi-laminar conditions when it impinges from below the set of spherical particles. This ping-pong ball on air table configuration is inspired by a previous work by Ojha et al., where the solution to the equation of movement of a Brownian particle (consisting of a ping-pong ball in an air table) was found and compared to their experimental results [12]. Fan power is carefully adjusted so that the particles never lose physical contact with the plate and so they keep rolling over the grid, much in the same way of the aforementioned and other previous works [12,54,56,57,59].

Within the appropriate ranges of fan power, air upflow past the spheres produces turbulent vortexes [12,56,61,62] that yield stochastic horizontal movement to the spheres and thus the particle dynamics (if sphere rolling is excluded) are strictly two dimensional. For a dimensional analysis with similar particles, please refer to the methods section in [12] and/or Appendix A in [56]. As fan power is increased, the system passes through a series of different physical configurations which are accessed through phase transitions. We have observed phase coexistence during these transitions for experiments in a range of values of particle density. We characterize particle density by means of the packing fraction, which is defined here as $\phi \equiv N\sigma^{2}/\left(4R^{2}\right)$, where *N* is the number of particles present in the system. For this set of experiments, we used 40≤N≤252, which roughly corresponds to packing fractions 0.12<ϕ<0.76. All curves presented in the manuscript result from averaging all of the present particles in each experiment.

We perform each experiment by setting the fan onto a constant power. Once a stationary state is achieved, we record the particles dynamics from above by means of a high-speed camera (Phantom VEO 410L) at 250 fps. We double check afterward that the registered interval of the experiment is in effect under steady-state conditions by plotting vs. time the relevant space-averaged magnitudes (e.g., average kinetic energy). In any case, it is rather straightforward to ensure steady-state conditions by waiting for a time interval equivalent to several collisions per particle [63]. Because the recorded stationary section of the experiment that is recorded is always 100 s long, a large set of steady-state statistical replicas corresponding to recorded frames is available to process (≈$2\times 10^{4}$. In this way, we can achieve statistical accuracy of our data. Data sets are obtained by processing experiment movies with a particle tracking code that we developed specifically for this configuration. This code is composed of a series of OpenCV [64] and TrackPy [65] functions, which allow to obtain all particle positions over the acquired images and tag each particle so that they will be tracked through the entire movie. An exact copy of the particle tracking code (in python language) is freely accessible [66].

As described in detail in a previous work (see Appendix A in [56]), we have achieved a high degree of accuracy in the particle tracking because measurement error of particle displacement between consecutive frames is not higher than 0.03σ (less than 3% relative to particle diameter).

### 2.2. Phase Behavior

Figure 2 presents a series of movie snapshots displaying the different phase states that we have detected in our experiments for two different packing fraction values (ϕ=0.18 for the first row, ϕ=0.55 for the second row). For each packing fraction, snapshots are placed in ascending order of fan power. For ϕ=0.18 and the lowest fan power, a subset of particles is still static, because the turbulent vortexes intensity is not strong enough so as to overcome static friction. We denote this static phase as *arrest* phase due to its static nature. It corresponds to the upper right corner in Figure 2a. Interestingly, the arrest phase has been detected before in analogous configurations [4] and is known to develop a quasi-static ordered state that has been denoted as *collapse* phase. If current intensity is high enough, however, a subset of the spheres can activate its thermal-like movement (that, as we discussed, is due to the turbulent vortexes generated by the upflow past the spheres). Their movement is initially limited so that we can observe caging effects for these particles (bottom left section in Figure 2a). Thus, we have detected coexistence between the arrest phase and a glass phase for the caged moving particles (Figure 2a). The arrest phase eventually disappears as particles gradually activate, giving rise to a pure glass phase and afterward (at stronger upflow intensity) to the coexistence of the glass–liquid phase (for this coexistence, see Figure 2b, with the glass phase occupying the lower-density region in the upper right corner of the snapshot). At higher density (ϕ=0.55, we observe consecutively: liquid phase (Figure 2c), liquid–crystal coexistence (the crystal is hexagonal, see the developing hexagonal structure, with some defects, in bottom right corner in Figure 2d), and crystal–liquid phase (Figure 2e). At this point, if the fan power is still increased, there is gradual shrinkage of the hexagonal crystal (which *melts*). The crystal completely disappears above a threshold value of air-current intensity. At this point, only the liquid remains (again). This last stage is not represented because they look much like the snapshots in Figure 2c,d. In any case, grasping the phase configuration out of these snapshots is not straightforward, and for this reason, we analyze in more detail the particle trajectory structure in the next section.

Please see, in the Data Availability statement, the web link where a movie version of Figure 2 can be downloaded.

## 3. Results

### 3.1. Trajectories and Granular Temperature Field

In order to analyze the dynamic properties (except for the static arrest phase) in more detail, we separately analyze, for each phase, the trajectory shape, the temperature field, and the properties of the diffusion and velocity autocorrelations. We define the temperature field as $T\left(x,y\right)=\left(1/2\right){\langle v^{2}\rangle }_{xy}$, where ${\langle v^{2}\rangle }_{xy}$ stands for the square of the particle velocity (*v*) averaged at a given point (x,y) of the system, through all the measured steady states. In this work, we use the concept of granular temperature (or simply, temperature) in the same sense as first defined by Kanatani [67].

Let us now comment on the phase behavior and the transitions we detected. In the first transition, at a low granular temperature, a glass is observed in coexistence with an arrest phase. By the arrest phase, we refer to particles that, at a very low-energy input, lie still due to friction [4,32]. In the second phase transitions, the glass decays to a liquid-like phase, coexisting with it as it shrinks. At even higher temperatures, a hexagonal crystallite develops in which we can observe, in certain ranges of the driving intensity, a coexistence between a liquid and a hexagonal crystal [56]. Moreover, we have detected that a strong energy non-equipartition occurs between the coexisting phases.

These results are illustrated in Figure 3. This figure shows, for a representative set of experiments, the particle trajectories in the left column, the 2D color maps of the granular temperature T(x,y) in the middle column, and the pair correlation function g(r) in the right column. The experiments shown in Figure 3 correspond to two different densities (low, with ϕ=0.18, and high, with ϕ=0.55), with each subset in an ascending order of the upflow-current intensity. In it, we can see the phases that consecutively appear as more energy is input into the system. Figure 3a shows two qualitatively different types of arrangements of particle-trajectory phases: a disordered lattice of caged particle trajectories (caged in the sense that the moving particles remain close to a disordered set of fixed points) and a disordered lattice of static particles (arrest phase). In effect, the former set of trajectories can be identified as a glass phase because, although the particles undergo a continuous stochastic movement, the caging effects are predominant [68,69] and a disordered but permanent particle trajectory structure (lattice) can be observed. To the best of our knowledge, it is not very common to find glass transitions at such low densities. With respect to the latter, it is apparent that the particles remain static during the complete 100 s experiment. From this qualitative difference between these two phases, a strong energy non-equipartition emerges. In effect, as we can see in Figure 3b, the region corresponding to the arrest phase has a vanishing granular temperature *T*, whereas for the glass phase *T* is clearly non-null. Note that, contrary to what has been observed in the thin layers, we have not detected a static phase that yields a hexagonal-ordered collapse phase, as in a vertically vibrated monolayer of spheres [4]. This peculiar arrest phase, which is also present in a vibrated granular monolayer [32], disappears here gradually as the upflow current increases, to a point where we can observe the two-phase coexistence between the glass-like and liquid-like phases, as in Figure 3c, where the liquid phase is observed in the region where all the trajectories mix and cross each other during the experiment; this is in contrast to the disordered pattern of the localized trajectories visible in the upper left corner. Figure 3d–f show a fully developed glass phase at a low density (ϕ=0.18). Figure 3g–i, at a higher temperature, show the glass-and-liquid phase coexistence. The g(r) curves for the glass phases (see the green curves in Figure 3c,f,i) do not help to distinguish the glass phase from the liquid phase. We will have to resort to the peculiar dynamic properties of the glass for that matter (the mean-squared displacement and velocity autocorrelations) [68]. As we can see in Figure 3e,h, the energy non-equipartition is strong here again, with the glass phase being noticeably cooler. At a higher density (packing fraction ϕ=0.55), we observe, consecutively, a monophase liquid-like system (Figure 3j–l); a developing hexagonal crystal which almost occupies the entire region (Figure 3m–o) and whose g(r) shows a higher first maximum; and a two-phase system, with a liquid coexisting with a hexagonal lattice (Figure 3r). With respect to the corresponding behavior of their pair correlation function, g(r), the curves for the liquid and the crystal are noticeably different, with the curves for the crystal having stronger secondary peaks, as usual [35]. With respect to T(x,y), notice also that the non-equipartition is also present in the case of the liquid–crystal two-phase system, with the crystal colder than the liquid (Figure 3h).

Overall, the fact that the non-equipartition is noticeably present in all the two-phase system configurations denotes that each phase has its own peculiar dynamics. In particular, this may be an indication that the diffusion process in each phase might have different scales and behavior [70]. In this sense, note that the structure of the Brownian trajectories in each phase is very different, as the trajectories in the left column of Figure 3 show. For this reason, by identifying first which trajectories belong to each phase in all the experiments (with the aid of the average coordination number for each particle, as extracted from the corresponding Voronoi tesselations [56]), we have computed the diffusion coefficient for each phase. This allows us to track the mean-squared displacement (MSD) for each phase independently. Figure 4 and Figure 5 show the evolution of the ensemble mean-squared displacements which in 2D can be defined as
(1)$$\langle \Delta r{\left(t\right)}^{2}\rangle \equiv \langle \Delta x{\left(t\right)}^{2}+\Delta y{\left(t\right)}^{2}\rangle ,$$
where $\Delta x{\left(t\right)}^{2}\equiv \left(1/N\left(t\right)\right)\sum_{\left\{t_{0}\right\}}{\left[x\left(t+t_{0}\right)-x\left(t_{0}\right)\right]}^{2}$ (and analogously for $\Delta y{\left(t\right)}^{2}$). For each lag time, and under steady-state conditions, the squared displacements $\Delta x{\left(t\right)}^{2},\Delta y{\left(t\right)}^{2}$ can be obtained from averages over the N(t) available initial times $t_{0}$ (basically, m where $N_{\mathrm{st},\mathrm{frames}}$ is the number of images taken under the steady-state conditions, in our case, $N_{\mathrm{st},\mathrm{frames}}≊$ 25,000). The diffusion coefficient can be obtained from the MSD, because most commonly the following relation is fulfilled [56,70]
(2)$$\langle \Delta r{\left(t\right)}^{2}\rangle =\left(4D\right)t^{\alpha },$$
where α is a constant usually called the diffusive exponent. (When Equation (Equation 2) is not fulfilled, the diffusion is said to be anomalous [70]). Trivially, from (Equation 2), the diffusive exponent α corresponds to the slope of the diffusive part, in a Log–Log representation, of the MSD vs. lag time, as in the curves displayed in Figure 4 and Figure 5. By the diffusive part, we mean, as usual, the part of the MSD vs. time curve that is after the ballistic regime, which should always have α=2 [70]. As a guide to the eye, the ballistic (α=2) and normal diffusion (α=1) values were indicated inside each panel in Figure 4 and Figure 5. However, we have found that the slope in the diffusive part of the MSD curves is not constant in some cases, and it may be said that the diffusion is anomalous in these cases, which is in agreement with the experimental observations in a previous analysis [59]. Therefore, the computed diffusive exponents α for these cases and the corresponding slope (and thus, diffusion coefficient) from Equation (Equation 2) represent an averaged value.

In Figure 4, at a low packing fraction (ϕ=0.18), we can see the MSD for the following cases: observed—glass (a), glass–liquid (b), and liquid (c); whereas, in Figure 5, we can see the cases only crystal (a), crystal–liquid (b), and only liquid (c). It is very apparent that the behavior of the MSD for each phase is very different. In particular, the monophase glass configuration (Figure 4a) presents an MSD with a local maximum at the end of the ballistic regime, after which it presents a characteristic curvature in the diffusive part of the curve, which is also strongly subdiffusive. The MSD behavior of the glass-like phase is thus characterized by a short plateau in the MSD followed by an increase (when the particles escape the current “caging” area and move to a new location). At a higher *T*, in Figure 4b, we can see the glass–liquid coexistence. In this case, the emerging liquid phase is still weakly subdiffusive (although with a clearly faster MSD, if compared to the companion glass). This can be attributed to a liquid structure that is still in development as the glass shrinks in size. In Figure 4c, the system with only a liquid phase already shows a normal diffusion scenario. In contrast, Figure 5a, at a higher packing fraction (ϕ=0.55), shows a single crystal configuration, with the diffusive part of the MSD close to stagnation (the zero time growth of the MSD), i.e., the dynamics are very strongly subdiffusive, as evidence of the crystalline diffusion [29]. In the case of the crystal–liquid coexistence, Figure 5b, the less disordered phase (glass) clearly undergoes subdiffusion, whereas the liquid has normal diffusion. Normal diffusion can also be seen in Figure 5c, where the single liquid phase is recovered. As we said before, and it is still worth remarking here again, an important and surprising result is the confirmation, in view of the MSD behavior illustrated in Figure 4, of a glass transition with clear caging processes at low densities, when in general these processes are observed (to the best of our knowledge) in dense granular fluids [71]. This result may be the outcome of an effective potential developed by the interaction between the spherical balls through the intermediate air flow. In the same way, the crystal appears at unusually low densities, in comparison with the case of hard particles [35].

### 3.2. Diffusion Coefficient

From the results in Figure 4 and Figure 5, we may conclude that the evolution of the MSD for each phase is qualitatively very different, which confirms our identification of the different observed phases, as previously discussed. Next, we computed the diffusion coefficient separately for each phase in this section. We represent in two figures our measurements of the diffusion coefficient. In Figure 6, *D* is represented vs. the packing fraction for a series of experiments in different ranges of *T*: $T/m<0.6\;\sigma^{2}/s^{2};\;0.6\;\sigma^{2}/s^{2}<T/m<0.8\;\sigma^{2}/s^{2};\;0.8\;\sigma^{2}/s^{2}<T/m<1.2\;\sigma^{2}/s^{2}$, and $T/m>1.2\;\sigma^{2}/s^{2}$, whereas in Figure 7, we plot *D* vs. *T* for three representative packing fraction values (ϕ=0.18;0.46;0.55).

Figure 6 highlights the diffusive stages of the different phase configurations, including those with a phase coexistence (the coexisting phases are here joined with dashed vertical lines). As we can see, at $T/m<0.6\;\sigma^{2}/s^{2}$ (top left panel), the diffusion coefficient generally tends to decrease with an increasing ϕ. Moreover, only glass or liquid phases are visible at a very low *T*, with the liquid coexisting with the glass at low packing fractions, whereas at intermediate packing fractions, we find the crystal–liquid coexistence and at a larger ϕ only the crystal is detected, in this case with the lowest *D* values. At higher intermediate temperatures (at $0.6\;\sigma^{2}/s^{2}<T/m<0.8\;\sigma^{2}/s^{2}$, in the top right panel, and at $0.8\;\sigma^{2}/s^{2}<T/m<1.2\;\sigma^{2}/s^{2}$, bottom left), we can see the glass–liquid at a low density and again the effects of a larger *T* cause the withdrawal of the crystal–liquid coexistence at an intermediate ϕ, leaving the liquid (red symbols) alone. Again, at a higher ϕ, the crystal–liquid and crystal are detected. Finally, in the largest range of values of *T*, it is apparent that only the liquid is observed (except for a configuration with the densest system we used) and that in this regime the diffusion coefficient is nearly constant with respect to the packing fraction, except for a steep decay at a large ϕ (where the only two cases of coexistence with a crystal are observed). It is also interesting to note that an extrapolation of the curve averaged by the crystalline states extends to the low-density glass transition zones. In summary, *D* tends to decrease for denser systems, except at a very high *T*, where it tends to keep approximately constant.

Now, Figure 7, which represents *D* vs. *T*, summarizes well the quantitative differences in the diffusion coefficient for the three phases (glass, liquid, and crystal), together with the ranges of the coexistence of the glass and crystal with the liquid phase. Overall, the liquid predominates at a low and a moderate density (the left and center panels), whereas the glass and crystal predominate at a very low and high density, respectively. It can also be observed that both the glass and crystal are less diffusive than the liquid, as it was to be expected, with the crystal having the lowest values, systematically, of the diffusion coefficient.

It is important to remark here that Figure 6 and Figure 7 show plenty of evidence of the finite differences in the diffusion coefficient in the same system state, as a strong experimental proof of the discontinuous phase transition and phase coexistence. Moreover, we have not observed in any case a continuously changing diffusion coefficient between the different phases.

As we mentioned before, previously to computing the diffusion coefficient, we determine the diffusive exponent α as defined by Equation (Equation 2), and whose value defines if the system is under superdiffusion (α>1), subdiffusion (α<1), or normal diffusion (α=1) [70,72]. So, we plot in Figure 8 the measurement of α for all the performed experiments altogether. They are represented as a function system granular temperature *T* for all the particle densities combined (here represented in the form of the packing fraction ϕ). The red points signal the liquid-phase diffusive exponents, green stands for the glass phase, and blue for the crystal. Note the logical order of α by the phases, with the crystal having the lowest values, the glass in the intermediate region, and the liquid having the largest α. Moreover, as we can see, the crystal and also the glass phases are very subdiffusive. The liquid, however, can be either weakly subdiffusive or weakly superdiffusive. The superdiffusive values (α>1) are reached after the phase coexistence has vanished, i.e., the pure-liquid phase tends to be superdiffusive, which we think is an indication again of the repulsive forces between the particles. Especially at not large densities, the repulsion between the particles may aide the particles diffuse in between the neighboring particles, thus enhancing the diffusion.

### 3.3. Velocity Autocorrelations

We also represented the velocity autocorrelation function, computing its trend for the glass, liquid, and crystal. The velocity autocorrelations provide information on the dynamics of the particle collision, in particular on the statistical relation between the pre-collisional and post-collisional velocities. We define the velocity autocorrelation function at lag time τ as usual [17]
(3)$$A_{v}\left(\tau \right)=\frac{\langle v(t)\cdot v(t+\tau )\rangle }{\langle v(t)\cdot v(t)\rangle },$$
where 〈⋯〉 stands for the ensemble averaging over all the steady states at initial times $t_{0}$. Figure 9 represents the velocity correlations $A_{v}\left(\tau \right)$ in the glass–liquid transition and Figure 10 represents $A_{v}\left(\tau \right)$ for the crystal–liquid transition. It is to be noted that the particles in the glass phase (the left panel in Figure 9) show strong velocity anticorrelations at early times ($A_{v}\left(\tau <1\right)<0$) and that these anticorrelations are transmitted to the coexisting liquid (the center panel of Figure 9). Surprisingly as well, the depth of the anticorrelation well is increased in the glass–liquid two-phase system, with respect to the pure glass (the left panel). Furthermore, the liquid remains anticorrelated at τ<1 even when the glass has disappeared at a high *T*. By contrast, the pure-liquid phase does not display the autocorrelations in the crystal–liquid transition (the right panel in Figure 10), as well as an increase in the time required for the autocorrelation function to cancel for the first time. However, there are weaker anticorrelations in the pure crystal (the left panel in Figure 10) and the crystal–liquid two-phase state, and a very short time for the first cancellation of the autocorrelation function, typical of the crystal phase. Let us remark here that the right panels in Figure 9 and Figure 10 combined reveal that the liquid phase has a variety of internal behaviors. This variety of behaviors is closely related to the occurrence of the glass transition at low densities because, as mentioned before, at low densities there is a repulsive interaction between the particles mediated by the upward air flow (as if they had a soft core with a diameter greater than that of the balls), and when the density is increased, this effective potential does not prevent the direct collision between the spherical balls.

## 4. Discussion

We have studied the nearly 2D dynamics of a system of (inelastic) rolling spheres under steady-state conditions. The dynamics of the set of spheres is sustained by the means of the turbulent vortexes that originate out of an air upflow past the spheres. As we have seen, the phase behavior of the system is very complex. We have been able to detect an arrest phase (the particles that remain still or static for the low-energy input), a glass phase (the disordered lattice of Brownian particles with sporadic jumps to other lattice positions), a liquid (a completely disordered phase), and a hexagonal crystal. The detection of each of the phases has been achieved by the means of a direct observation of the arrangement of the particles and/or an analysis of the properties of the corresponding pair correlation function. Interestingly, the glass phase appears at very low densities, which is a very rare situation [69]. The appearance of this low-density glass is undoubtedly due to the underlying long-ranged repulsive forces between the air-fluidized particles [54]. However, we need to point out that one could not expect that the existence of these long-ranged interactions would necessarily lead to the emergence of a low-density glass [55]. As a matter of fact, this kind of low-density glass phase had not been reported previously in granular matter, to the best of our knowledge. Moreover, the glass and the hexagonal crystal can coexist with the liquid (again, the glass–liquid coexistence had not been reported previously in the context of granular dynamics, to the best of our knowledge).

Additionally, we have observed that the glass can also coexist with the arrest phase, at a low air-current intensity (the dynamics in the process of activation were again not detected previously). In fact, the dynamics of the system are so complex that we have been able to detect important qualitative differences in the behavior of a single phase. For instance, as we mentioned above, the velocities of the particles in the liquid phase can be either strongly anticorrelated at early times or not anticorrelated at all, depending on the configuration of the system. As another example, the crystal can display a vanishing diffusive exponent (and in any case, the crystal is always strongly subdiffusive; see Figure 8).

For all the coexisting phases, we have observed a strong energy non-equipartition and finite jumps in the values of magnitudes, such as the diffusion coefficient (and its diffusive exponent; see Figure 6, Figure 7 and Figure 8), and the depth of the first minimum of the respective velocity autocorrelation function. This supports the evidence that the phase transitions we described for these experiments are discontinuous, as opposed to the continuous nature of the melting transition described in the KTHNY scenario. Furthermore, the diffusive properties of the observed phases are, in general, rather different. Both the glass and the crystal are always subdiffusive and present anticorrelated velocities. In contrast, the liquid presents an always nearly normal diffusion. Furthermore, all the transitions observed in this work occur through the phase coexistence, contrary to the liquid–hexatic–crystal continuous phase transition without the coexistence that occurs for the KTHNY scenario. Specifically, the fact that the hexagonal crystal in our setup melts by shrinking in size, giving rise to a growing liquid, and that the transition is not defect mediated, guarantees the complete absence of a hexatic phase and a defect-mediated melting process.

Our experimental configurations are very far away from equilibrium because they simultaneously undergo strong repulsive forces (due to the interstitial air) and energy dissipation (due to friction). We think this is the reason for the strong differences between the phase transition scenario observed for air-fluidized rolling spheres and the KTHNY scenario described for both 2D equilibrium systems [45] and (under certain conditions) non-equilibrium systems, such as a vibrated granular monolayer [35,46] and 2D active Brownian disks [53]. Here, however, this KTHNY scenario is, as we said, absent and the hexatic phase has not been observed in any situation, which is a very peculiar situation in the context of two-dimensional matter. In this sense, the results reported here can provide more insight for the development of the theory on 2D phase transitions for non-equilibrium systems.

It remains for future theoretical work to study in more detail the structure of this intriguing phase behavior.

## Figures and Tables

**Figure 1 entropy-24-01684-f001:**
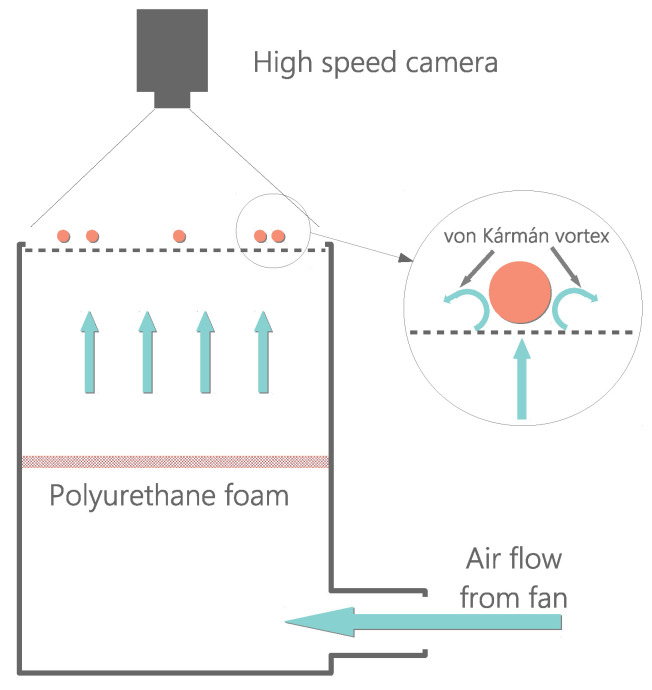
Sketch of the experimental set up.

**Figure 2 entropy-24-01684-f002:**
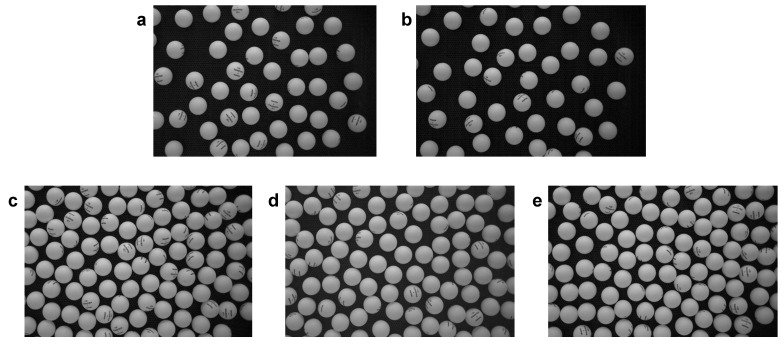
Snapshots of the different phase configurations observed in experiments. Packing fraction is ϕ=0.18 for (**a**,**b**) and ϕ=0.55 for (**c**–**e**). Granular temperatures are, in order: $T/m=\left[0.16,\;0.74,\;0.38,\;0.47,\;0.70\right]\;\sigma^{2}s^{-2}$ for the configurations (same order): glass–arrest phase, glass–liquid, liquid, crystal, crystal–lquid.

**Figure 3 entropy-24-01684-f003:**
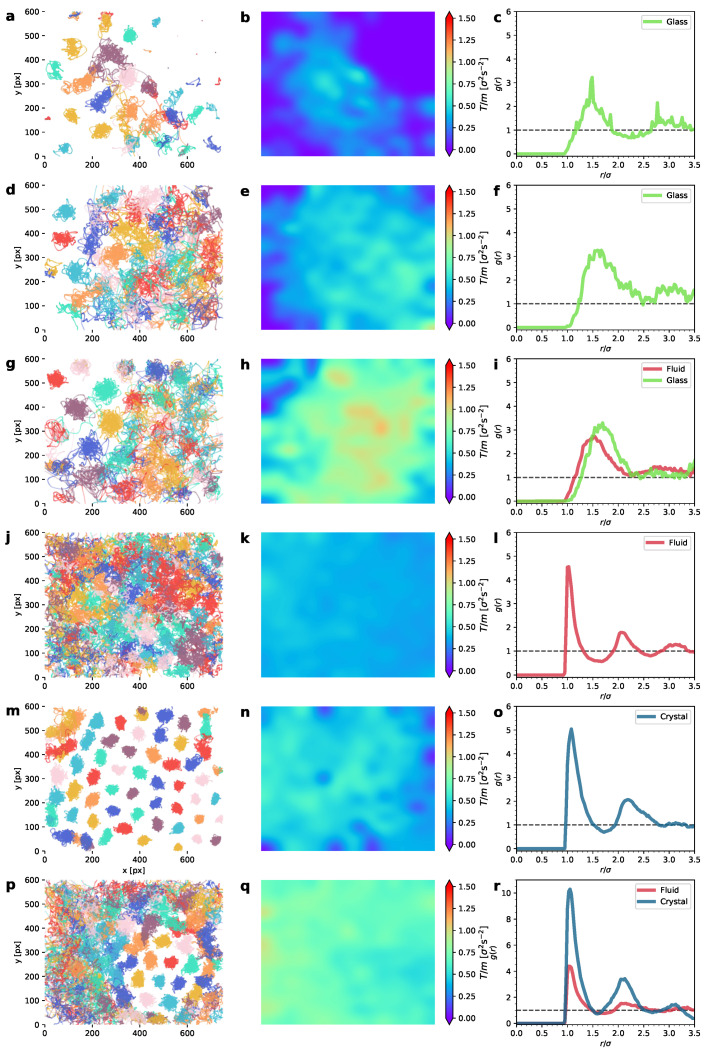
Phase behavior of our system in a central region of interest. Left column represents particle trajectories; right column shows the corresponding granular temperature 2D fields (*T*). Packing fraction is ϕ=0.18 for (**a**–**f**) and ϕ=0.55 for (**g**–**o**). The values of the granular temperature for each column of panels are given by, from top to bottom, respectively: $T/m=\left[0.16,\;0.69,\;0.74,\;0.38,\;0.47,\;0.70\right]\;\sigma^{2}s^{-2}$. In (**a**–**c**), we can see phase coexistence between a glass phase and the arrest phase at low density. In (**d**–**f**), low density but higher *T*, there is only glass, whereas for (**g**–**i**) there is glass-and-liquid phase coexistence. (**j**–**l**) shows that the system is completely disordered (there is only a liquid phase) state that can be observed at intermediate temperatures for all densities. At higher densities, if the liquid is further heated (air upflow is increased), a cooler crystallite develops in coexistence with the liquid; the crystal grows as *T* is increased, eventually occupying the entire system, as in (**m**–**o**). At stronger driving, the crystal tends to melt, thus shrinking in size, as seen in (**p**–**r**).

**Figure 4 entropy-24-01684-f004:**
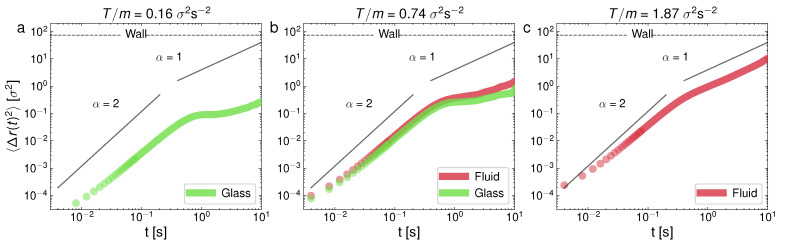
Log–Log representation of the mean-squared displacement for three representative experiments of ϕ=0.18, for different *T*: (**a**) $T/m=0.16\;\sigma^{2}s^{-2}$; (**b**) $T/m=0.74\;\sigma^{2}s^{-2}$; (**c**) $T/m=1.87\;\sigma^{2}s^{-2}$. Case (**b**) corresponds to the state shown in Figure 3c,d.

**Figure 5 entropy-24-01684-f005:**
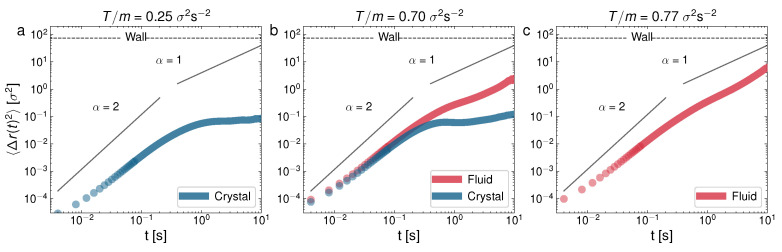
Log–Log representation of the mean-squared displacement for three representative experiments of ϕ=0.55, for different *T*: (**a**) $T/m=0.25\;\sigma^{2}s^{-2}$; (**b**) $T/m=0.70\;\sigma^{2}s^{-2}$; (**c**) $T/m=0.77\;\sigma^{2}s^{-2}$. Case (**b**) corresponds to the state shown in Figure 3i,j.

**Figure 6 entropy-24-01684-f006:**
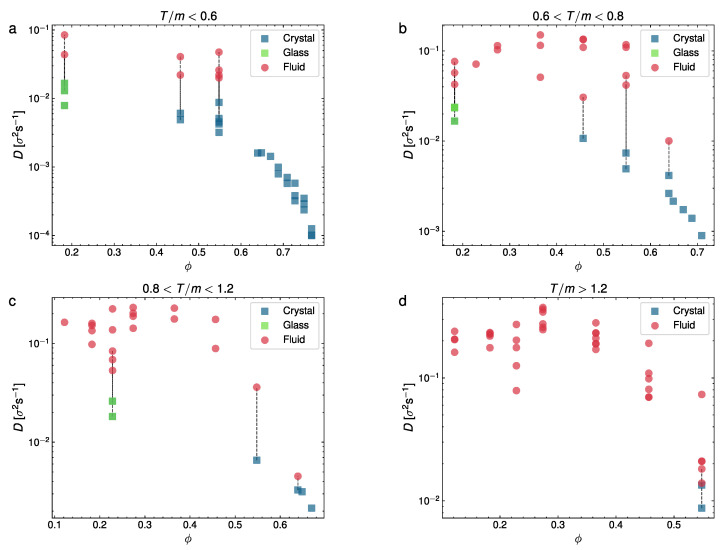
Diffusion coefficient *D* vs. packing fraction ϕ divided in four panels by the overall granular temperature of each experiment, for different temperature ranges (in units of $\;\sigma^{2}s^{-2}$): (**a**) T/m<0.6, (**b**) 0.6<T/m<0.8; (**c**) 0.8<T/m<1.2; (**d**) T/m>1.2. Each point corresponds to an experiment. Wherever coexistence is visible, we have split *D* into two different points for the fluid (red) and crystal/glass phase (blue/green).

**Figure 7 entropy-24-01684-f007:**
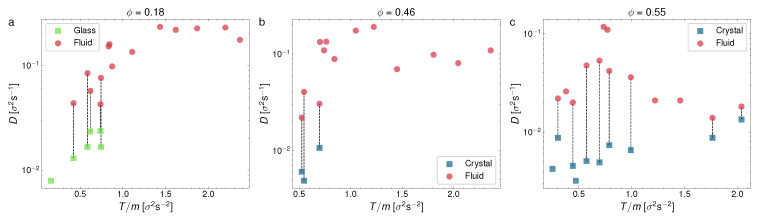
Average diffusion coefficients represented against granular temperature for three different packing fractions: (**a**) ϕ=0.18; (**b**) ϕ=0.46; (**c**) ϕ=0.55. Each point corresponds to an experiment. Wherever coexistence is visible, we have split *D* into two different points for the fluid (red) and crystal/glass phase (blue/green).

**Figure 8 entropy-24-01684-f008:**
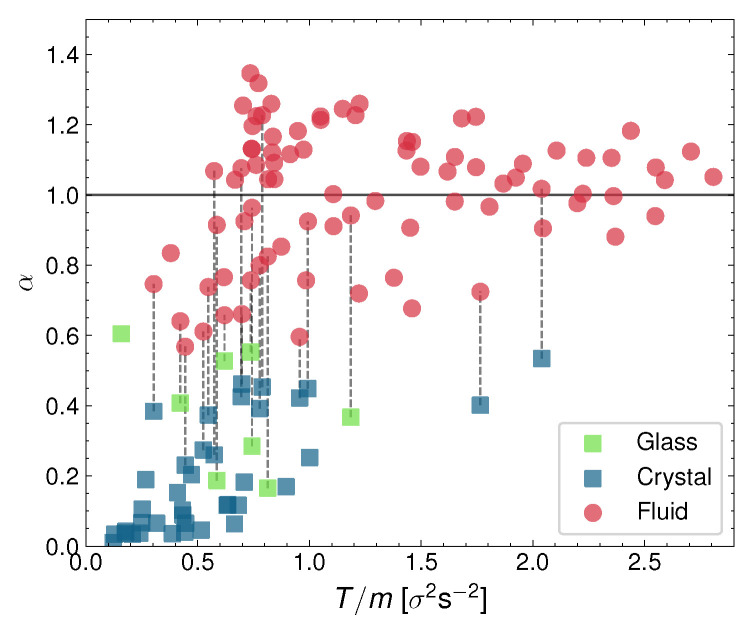
Diffusive exponent represented against granular temperature for all experiments. It has been calculated by averaging the logarithmic slope of the MSD in the [3–6] s range. Each point corresponds to an experiment; where coexistence is visible, we have split *D* into two different points for the fluid (red) and crystal/glass phase (blue/green).

**Figure 9 entropy-24-01684-f009:**
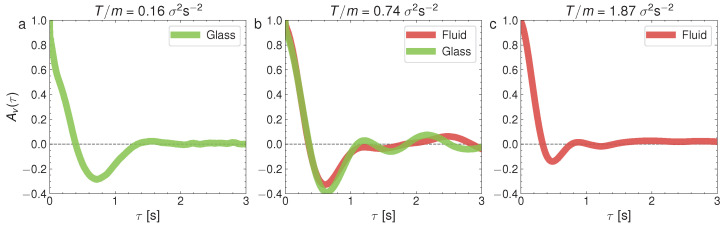
Normalized velocity autocorrelation at ϕ=0.18, and for three different values of *T*: (**a**) $T/m=0.16\;\sigma^{2}s^{-2}$; (**b**) $T/m=0.74\;\sigma^{2}s^{-2}$; (**c**) $T/m=1.87\;\sigma^{2}s^{-2}$. They correspond to the cases presented for the MSD in Figure 4.

**Figure 10 entropy-24-01684-f010:**
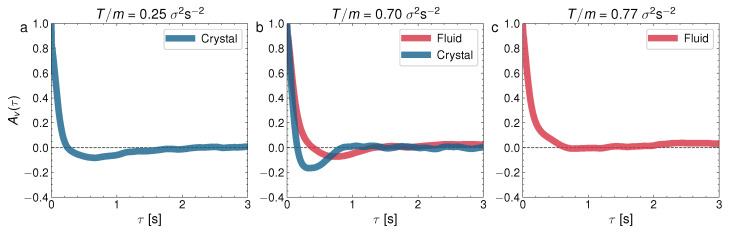
Normalized velocity autocorrelation for ϕ=0.55, and three different *T*: (**a**) $T/m=0.25\;\sigma^{2}s^{-2}$; (**b**) $T/m=0.70\;\sigma^{2}s^{-2}$; (**c**) $T/m=0.77\;\sigma^{2}s^{-2}$. They correspond to the cases presented for the MSD in Figure 5.

## Data Availability

The experimental data tables, trajectories, and movie corresponding to Figure 2 are available in a public repository at: https://doi.org/10.5281/zenodo.7264865, accessed on 10 November 2022.

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
