# Peer review of "Diffusion and Velocity Correlations of the Phase Transitions in a System of Macroscopic Rolling Spheres"

_entropy, 2022, doi:10.3390/e24111684_

Round 1
Reviewer 2 Report
In their manuscript, Vega Reyes et al study a granular monolayer of plastic spheres fluidized by an upwards air flow, which they use to study macroscopic phase transitions and coexistences using high-speed video recordings. The authors report and characterize two instances of macroscopic two-phase coexistence. They measure the granular temperature, mean square displacements, generalized diffusion coefficients, and velocity autocorrelation in the various phases. They claim, in particular, to identify a glass phase at unusually low packing fraction of particles.
In this paper, at least 10 out of 41 (~25%) of the references are self-citations, which seems unusually high.
The English and the general structure of this manuscript are fine, although have room for improvement.
Importantly, my general impression is that this manuscript substantially lacks in scientific rigor. For instance, it includes inaccurate—or lacks—definitions of key physical observables; apparent misunderstanding of statistical concepts; numerous vague statements; over-stated claims; an strikingly uninformative figure (Figure 2).
Please find below more detailed comments.
Introduction
- l.38 - “As it is well known, an equilibrium fluid in two-dimensions (2D) crystallizes to a hexagonal phase (the hexatic phase) via a continuous transition that is mediated by a phase that is specific of two dimensions.”
This statement is incorrect: crystalline and hexatic phases are precisely distinct (notably through the range of their respective translational order, i.e., quasi-long range (crystalline) vs short range (hexatic)). The hexatic phase is the intermediate phase between the solid (hexagonal) and isotropic liquid phase.
- l.42 - The hexatic phase is not “new” since it was predicted more than 40 years ago by the authors cited in the manuscript (with a typo in the first author’s name).
- l.42 - “The new hexatic phase appears exclusively in 2D and is characterized by having quasi-long-ranged translational correlations (with power-law decay) and short-ranged translational correlations (with exponential decay).”
There is a major error/ typo here: “quasi-long-ranged translational correlations” should be “quasi-long-ranged orientational correlations.” Defining incorrectly the properties of the hexatic phase, a well-known result to which the authors refer again in the conclusion, is very concerning to me. This and other mistakes make me question the general attention paid to writing this paper.
2. Description of the experiments
- How laterally homogeneous is the “homogenized upflow” between the polyurethane foam and the metallic grid? There is no characterization or reference, and this is an extremely important information to interpret the results presented in this manuscript.
- How laminar is the air flow past the 3 mm holes of the metallic grid? Were it not laminar, I believe it would jeopardize the idea the said turbulent vortices are formed as illustrated. Additionally, I find insufficient — if not inappropriate — to cite “An Album of Fluid Motion”, which is a collection of photographs, to support the statement that “von Karman vortices” are formed around each particle. A proper reference is needed.
- l.107 - The packing fraction is incorrectly defined, since \sigma was earlier defined as the diameter of the spheres, not their radius.
- “Everything is recorded”
This is unacceptably vague. The authors should make more efforts to be more specific here and throughout their manuscript.
- l.114 - “much higher than the typical transitory time towards the steady state.”
What is the typical transitory time towards the steady state?
- l.114 - “In this way, a set of 2 x 10^4 steady state statistical replica (corresponding to recorded frames after the transient) are available to process”
Calling the “recorded frames after the transient” “statistical replica” is incorrect. Replicas would be obtained from independent iterations of an experiment under the same set of conditions. As described here, the experiment was ran once, then extensive recordings were made. Therefore there is a single iteration of the experiment. If the authors want N replicas, then they need to re-initialize their experiment between each run, and carry out and record the resulting steady states of N such runs.
- l.127 - What data is there to back up the claim “At sufficiently high air current intensity, the vortexes become strong enough so that all particles undergo stochastic movement.”? A PDF of velocities could help support that claim.
- l.129-138/ Figure 2 - Figure 2 is strikingly non-informative, as the author partially admit themselves (l.126 - “In any case, grasping the phase configuration out of these snapshots is not straightforward”). There is no noticeable relevant difference between panels a and b, and between panels c, d, and e. In my opinion, this figure thus has no point being included, at least as such. Particle trajectories, and/or velocity arrows, etc., could be added to make the figure useful to the reader. Finally, I do not see any “crystalline” region in panels d and e. If there is an actual crystalline phase forming indeed, this is not a relevant illustration of it.
- Granular temperature: it appears in the figures from Figure 2 and is used throughout the manuscript, but it defined nowhere, which should clearly be addressed since it is a key physical observable in this manuscript.
3. Results
- Figure 3 — Lacking scalebars or x- and y-axes values. The reader cannot unambiguously understand what this figure is showing otherwise.
- l.141 “In order to analyze in more detail the dynamic properties”
No quantitative detail has been given to the reader yet, thus “more” may be removed.
- l.151 “In effect, the former set of trajectories can be identified as a glassy phase since, although particles undergo continuous stochastic movement, caging effects are predominant.”
I am questioning the accuracy of this statement. Caging, in a glass phase, occurs due to interaction between particles. But here, what are the arguments demonstrating that the observed “caging” is due to particle-particle interactions rather than, more trivially, resulting from friction under the condition of low kinetic energy injection?
l.154 “it is not very common to find glass transitions at such low densities”
I think that this key statement is misleading. The relevant density is the effective density of the particles with their interaction potential, not the density of the plastic spheres themselves. The authors claim later (e.g., l.281) that these particles can be considered as being “soft”, “with a diameter greater than that of the balls.” I strongly suspect that the density of those larger, soft effective particles would be much closer to the densities at which the glass transition is usually observed.
l.203 - “An important and surprising result is the observation of glassy transitions with clear caging processes at low densities, when in general these processes are observed (to the best of our knowledge) in dense granular fluids” Again, I strongly believe that an effective density is the relevant observable here.
l.206 - “This result may be the outcome of an effective potential developed by the interaction between the spherical balls through the intermediate air flow.” The author should provide some quantitative argument to back up this suggestion, otherwise this remains unconvincing.
l.210 - “we compute the diffusion coefficient” In fact, what the author compute is the “generalized diffusion coefficient,” since diffusion here is anomalous.
Figure 4 + onwards
- How are particles classified in one class of another? This is not specified, although classification is a key element of the discussion.
Figure 4
- The main text refers to panel labels, but the figure includes no panel label
Figure 5
- Labelling as “crystal” particles that display non-vanishing diffusion (\alpha > 0) in the post-ballistic regime (t > 1s) is a problem. In particular, “crystal” particles in the central panel show very similar MSDs as those labeled “glass” in the central panel of Fig. 4. What is the criteria to classify some particles as “crystal” vs "glass"?
Conclusion
- “The dynamics of the set of spheres is activated by means of the turbulent vortexes that originate out of an air upflow past the spheres”
Again, what is the basis for this statement?
- “this KTHNY scenario is absent and the hexatic phase has not been observed”
Again, the authors give no proof that the solid phase they find is in fact crystalline (e.g., using pair correlation function or structure factor).
Reviewer 3 Report
The authors report the results of experiments of a monoloayer of spheres levitated by a suitable current of air. A rich phenomenology of phases and their associated transitions is described.
The manuscript presents results that are interesting enough, however there are several points that need to be addressed before considering publication:
11. In the introduction the authors discuss very briefly some aspects of granular matter. Granular media are considered those in which particles interact via contacts. However, in the present case, as the authors note, the air induces an effective potential that mediates the interaction of the spheres. Thus, a more detailed discussion is required regarding how similar or different the present system is to a (dry) granular system
22. Related to the above, in which cases the air is more important than the contacts between the spheres? Is there a regime of collisional vs. long-distance interaction depending on packing fraction or granular temperature? The authors only make one comment in lines 206-207.
33. The structure of the monolayer should be assessed by the radial distribution function. Claims regarding crystal vs. glass, not to mention the more specific hexagonal crystal are not supported by the evidence given in the manuscript.
44. The discussion regarding diffusion starting in line 301 is lacking and puzzling. First, there is no attempt of explanation of the switching super/sub-diffusive behavior of the liquid phase. Moreover, the authors say that values departing from normal diffusion are due to limitations of the experimental apparatus or errors. If there are unreliable points, they should be eliminated from the final data. If the final data still contains evidence of this switching super/sub-diffusive behavior, it requires a more detailed discussion. There are a number of reasons to observe super- or sub-diffusion, such as Lévy flights or long waiting times, have the authors found any of these or other reported in literature?
55. The definition of granular temperature should be given, as is done with the other calculated quantities.
66. How are the maps of granular temperature (Fig.3) obtained? Is there an averaging, smoothing? And how precisely is it done?
77. How many spheres are used in the experiment and, more important, how many are used for the calculations reported?
88. As a general comment, I find that the authors describe their findings more than trying to grasp the subjacent reasons. This is specially so in the final section. This aspect of the manuscript should be improved.
99. Least but not last, it must be explained how the present manuscript adds or improves the analysis of the same experiment presented in Ref. 28 by the same authors.
Round 2
Reviewer 2 Report
To the Authors and the Editors.
In my first report, my overall recommendation was to reject the paper, for the reasons that I carefully detailed in that report. If, in contrast, I had concluded that 'Reconsider after major revision' was a more appropriate recommendation, I would have chosen it at that time-- which I did not. Therefore, my recommendation remains to reject the paper.
Reviewer 3 Report
I thank the authors for answering my questions and comments. I have no further remarks.